# Strand-Specific RNA Sequencing Reveals Gene Expression Patterns in F1 Chick Breast Muscle and Liver after Hatching

**DOI:** 10.3390/ani14091335

**Published:** 2024-04-29

**Authors:** Jianfei Zhao, Meiying Chen, Zhengwei Luo, Pengxin Cui, Peng Ren, Ye Wang

**Affiliations:** 1School of Life Science and Engineering, Southwest University of Science and Technology, Mianyang 621010, China; zhaojf@swust.edu.cn (J.Z.); cmy18599950531@163.com (M.C.); 13198691829@163.com (Z.L.); 15138181151@163.com (P.C.); 2Sichuan Key Laboratory of Conservation Biology on Endangered Wildlife, Chengdu Research Base of Giant Panda Breeding, Chengdu 610081, China

**Keywords:** gene expression patterns, transcriptome, heterosis, chicken, additivity

## Abstract

**Simple Summary:**

Understanding post-hatch gene expression patterns is crucial for exploring the genetic basis underlying economically important traits in the crossbreeding of chickens, which has been rarely studied. Therefore, we conducted gene expression analysis on F1 chicken breast muscle and liver tissues using ssRNA-seq at 28 days. The study revealed additivity as the predominant gene expression pattern in post-hatch muscle and liver. GO analysis identified 11 biological process terms associated with growth and development in differentially expressed gene sets and non-additive gene sets, including key genes like *STAT5A* and *TGFB2*. KEGG analysis uncovered six growth-related pathways with genes such as *SLC27A4* and *GLUL*. These findings provide valuable insights for domestic animal crossbreeding.

**Abstract:**

Heterosis refers to the phenomenon where hybrids exhibit superior performance compared to the parental phenotypes and has been widely utilized in crossbreeding programs for animals and crops, yet the molecular mechanisms underlying this phenomenon remain enigmatic. A better understanding of the gene expression patterns in post-hatch chickens is very important for exploring the genetic basis underlying economically important traits in the crossbreeding of chickens. In this study, breast muscle and liver tissues (*n* = 36) from full-sib F1 birds and their parental pure lines were selected to identify gene expression patterns and differentially expressed genes (DEGs) at 28 days of age by strand-specific RNA sequencing (ssRNA-seq). This study indicates that additivity is the predominant gene expression pattern in the F1 chicken post-hatch breast muscle (80.6% genes with additivity) and liver (94.2% genes with additivity). In breast muscle, Gene Ontology (GO) enrichment analysis revealed that a total of 11 biological process (BP) terms closely associated with growth and development were annotated in the identified DEG sets and non-additive gene sets, including *STAT5A*, *TGFB2*, *FGF1*, *IGF2*, *DMA*, *FGF16*, *FGF12*, *STAC3*, *GSK3A*, and *GRB2*. Kyoto Encyclopedia of Genes and Genomes (KEGG) annotation presented that a total of six growth- and development-related pathways were identified, involving key genes such as *SLC27A4*, *GLUL*, *TGFB2*, *COX17*, and *GSK3A*, including the PPAR signaling pathway, TGF-beta signaling pathway, and mTOR signaling pathway. Our results may provide a theoretical basis for crossbreeding in domestic animals.

## 1. Introduction

As the most numerous and ubiquitous domestic animal, chickens (Gallus gallus domesticus) provide abundant, high-quality, and affordable meat and egg protein for human production and livelihood [1]. During the 20th century, specialized meat- and egg-type chickens were established separately to avoid the inherent conflict in selecting for both growth and reproductive traits in the same bird [2]. Due to the huge genetic variations and phenotypic trait differences between highly selected layer and broiler breeds [3], as well as the characteristics of a small genome, high reproductive performance, short generation intervals, and low feeding costs, chickens have become an ideal model for crossbreeding studies in domesticated animals.

Heterosis, also known as hybrid vigor, refers to the phenomenon where the survival or performance of a hybrid offspring is superior to the average of its genetically distinct parents [4]. Therefore, crossbreeding for heterosis has become an efficient strategy and has led to huge improvements in performance or adaptability in both crops [5,6] and livestock [7,8]. Previously, dominance [9], over-dominance [10], and epistasis [11,12] have been the classical theoretical hypotheses proposed to explain the genetic mechanism underlying heterosis. With the advancement of sequencing technology, omics studies such as transcriptomics, proteomics, and metabolomics have provided us with new insights into epigenetic gene expression and network changes, thereby deepening our understanding of the molecular mechanisms underlying heterosis [13,14].

Gene expression can be viewed as an intermediate phenotype that connects the genotype and specific traits [15]. Transcriptomic studies on crossbreeding have been reported in various species, including maize [16], rice [17], Arabidopsis [18], pufferfish [19], silkworms [20], pigs [21], and sheep [22], indicating that different factors such as species, hybrid combinations, and gender can influence the gene expression patterns between the parents and offspring. In chickens, additivity was identified as the predominant gene expression pattern in F1 chicken (Fayoumi × Leghorn) embryonic brain and liver by RNA-seq analysis [23]. However, due to the differences in breeds and the spatiotemporal specificity of gene expression, this study was unable to comprehensively and effectively reveal the gene expression patterns of economic traits in post-hatch chickens.

Previously, a full-sib F1 population of meat- and egg-type chickens were established to explore allele-specific expression and allelic transmission ratio distortion by whole-genome resequencing and strand-specific RNA sequencing (ssRNA-seq) [24,25]. The findings indicate that the phenomenon of allelic functional differences is commonly present in chickens and is crucial for the formation of the economic trait [26]. Specifically, ssRNA-seq can improve the accuracy of transcript quantification by reducing the proportion of read ambiguity compared with traditional RNA-Seq [27]. Herein, the previously ssRNA-seq data from the breast muscle and liver tissues of this full-sib F1 hybrid at 28 days of age, along with additional newly generated data from the parental pure lines, will be further used to explore the gene expression patterns in post-hatch skeletal muscle of chickens. The aim of this study was to explore the gene expression patterns of economically important traits and perform functional enrichment analysis to uncover crucial pathways or genes that may lead to heterosis in chickens. The results may provide a theoretical foundation for animal crossbreeding.

## 2. Materials and Methods

### 2.1. Experimental Design and Sample Collection

As described previously, one male meat-type chicken (Recessive White chicken) and a female egg-type chicken (Lohmann pink layer) were used to establish a full-sib F1 population [24]. Based on previous studies on growth rate and transcriptome sequencing data, we observed a significant turning point and differences in the growth rate phenotype and number of differentially expressed genes of the hybrid F1 population at 28 days compared to other time points [26]. Therefore, ssRNA-seq data from day 28 breast muscle and liver samples of the parental lines and the F1 cross were further used to explore the gene expression patterns (Figure 1). All birds were raised under the same ambient conditions with free access to feed and water. At the age of 28 days, we randomly selected 6 birds in the paternal line, maternal line, and F1 cross to collect breast muscle and liver tissues. All samples were immediately frozen in liquid nitrogen and stored at −80 °C until RNA extraction.

### 2.2. RNA Isolation and ssRNA-Seq

RNA isolation and ssRNA-seq were performed as described previously [24]. Briefly, total RNA was extracted from 36 tissue samples using RNA simple Total RNA Kit (TIANGEN Biotech, Beijing, China) according to the manufacturer’s protocols. Subsequently, the purity and concentration of total RNA were determined by a NanoDrop 2000C Spectrophotometer (Thermo Fisher Scientific, Waltham, MA, USA), a Bioanalyzer 2100 (Agilent Technologies, Santa Clara, CA, USA), and agarose gel electrophoresis to ensure that RNA samples met the requirements. The strand-specific cDNA library was constructed following the “Directional mRNA-Seq Library Prep Pre-Release” protocol by Illumina [28] and then was paired-end sequenced (2 × 150 bp) on the Illumina HiSeq X Ten System (San Diego, CA, USA) at Beijing BioMarker Biotechnology Co., Ltd., Beijing, China.

### 2.3. Transcriptomic Analysis

The transcriptomic analysis was conducted according to our previous study [29]. The chicken reference genome (Gallus_gallus-5.0.91) and corresponding gene annotation file were obtained from NCBI. Transcript quantification was performed using Salmon software version 1.3.0. Differential gene expression analysis was conducted using the edgeR package in R software 4.2.1, with false discovery rate (FDR) adjustment for multiple testing. Significance was defined as the FDR adjusted *p*-value < 0.05 and the absolute value of the log2FoldChange > 1. Principal component analysis (PCA) was conducted, and the results were visualized using R software. Enrichment analysis of differential genes was annotated using the DAVID online tool (https://david.abcc.ncifcrf.gov, accessed on 17 March 2024) with the Gene Ontology (GO) and Kyoto Encyclopedia of Genes and Genomes (KEGG) databases. An online platform (https://www.bioinformatics.com.cn, accessed on 19 March 2024) was chosen for data analysis and visualization [30].

### 2.4. Gene Expression Pattern Analysis

Subsequently, the transcriptomic data from 36 individuals (18 liver samples and 18 breast muscle samples) were further applied to analyze the gene expression patterns in the breast muscle and liver tissues of this cross population at 28 days of age. In this study, we established a criterion where genes expressed in at least 50% or more individuals within each tissue (out of the total 18 samples) were selected for subsequent analysis of gene expression patterns. The mid-parent gene expression values (MPVs) were calculated by taking the means of normalized gene counts from the paternal lines. Differentially expressed genes between the cross and MPV were identified using the *t*-test, and subsequently, the Benjamini/Yekutieli multiple correction method was applied to adjust for multiple comparisons.

Herein, genes that did not show significant differences between the cross and MPV were classified as exhibiting additive expression, while genes with significant differences were defined as non-additive (over-dominance, under-dominance, and dominance) genes (Figure 1) [23]. Specifically, the over-dominance and under-dominance expression patterns were determined based on whether the gene expression in the cross was significantly higher than that in the high parent or lower than that in the low parent, respectively. Furthermore, the dominance expression pattern can be further categorized into enhancing dominance (where there is no significant difference in gene expression between the cross and the high parent, but a significant difference compared to the low parent) and suppressing dominance (where there is no significant difference in gene expression between the offspring and the low parent, but a significant difference compared to the high parent). Finally, the gene sets corresponding to each gene expression pattern were further analyzed using the DAVID online tool mentioned above. In the present study, analysis was performed with homemade scripts when not specified.

## 3. Results

### 3.1. Analysis of Sequencing Data

In this study, a total of 149.97 Gb sequencing data (Q30 > 85%) were obtained from 24 samples of the parental lines, comprising 12 breast muscle tissues and 12 liver tissues (Appendix A). Meanwhile, the ssRNA-seq data obtained from the breast muscle and liver tissues of 28-day-old F1 birds in our previous study [28] were specifically selected for analyzing the gene expression patterns in F1 hybrids of meat-type and egg-type chickens. The PCA results demonstrated that breast muscle formed a distinct cluster set apart from liver, while no clearly distinct clusters were observed between different groups (Appendix A).

### 3.2. Top 10 Genes in Breast Muscle and Liver Expression Profile

By summing and sorting the expression of each transcript in the breast muscle and liver tissues of each individual in the paternal line, maternal line, and F1 hybrids, we found that the top 10 transcripts with the highest expression levels in the breast muscle corresponded to *GAPDH*, *ENSGALG00010010295*, *ENSGALG00010000588*, *MYLPF*, *ENSGALG00010029554*, *COX1*, *COX2*, *ATP6*, *COX3*, and *TPI1* (Figure 2A,B). The top 10 genes in the liver were *ALB*, *ENSGALG00010000588*, *COX1*, *COX2*, *ATP6*, *COX3*, *PIT54*, *HPX*, *CYTB*, and *APOA1* (Figure 2C,D). Additionally, *ENSGALG00010000588*, *COX1*, *COX2*, *ATP6*, and *COX3* were highly expressed in both breast muscle and liver. In particular, the mRNA level of *ENSGALG00010000588* in both breast muscle and liver tissues of the F1 cross was significantly lower than that in both the paternal line and maternal line. Furthermore, the *PIT54* gene in the liver showed a similar expression trend. In both tissues, no significant difference was observed in the expression levels of these genes between males and females within the same group.

### 3.3. Analysis of Differentially Expressed Genes

Differentially expressed genes (DEGs) between different groups (maternal line vs. F1 cross, paternal line vs. F1 cross, paternal line vs. maternal line) of the breast muscle and liver tissues were analyzed (Figure 3A). The number of DEGs of maternal line vs. F1 cross, paternal line vs. F1 cross, and paternal line vs. maternal line in the breast muscle was 1206, 1126, and 758, respectively, with 17 DEGs being owned by all combinations (Figure 3B). Moreover, the number of DEGs of the different comparison groups above in the liver was 1751, 920, and 469, respectively (Figure 3C), with 54 DEGs being shared by them.

Meanwhile, the upregulation and downregulation of these DEGs were further analyzed. The results revealed that in the liver, there were 300 upregulated and 169 downregulated DEGs in the comparison between the paternal line and maternal line (Figure 4A), 610 upregulated and 310 downregulated DEGs in the comparison between the paternal line and F1 cross (Figure 4B), and 902 upregulated and 849 downregulated DEGs in the comparison between the maternal line and F1 cross (Figure 4C). In addition, our results showed that the numbers of upregulated and downregulated DEGs of the paternal line vs. maternal line, paternal line vs. F1 cross, and maternal line vs. F1 cross in breast muscle were 475 and 283 (Figure 4D), 491 and 635 (Figure 4E), and 481 and 725 (Figure 4F).

### 3.4. Analysis of Gene Expression Pattern

In this study, a total of 15,496 candidate genes in the liver tissue and 14,299 candidate genes in the breast muscle tissue were detected for subsequent gene expression pattern analysis, based on their expression in more than 50% of individuals within each tissue. There were 3002 genes in the liver and 831 genes in the breast muscle expressed differentially between the F1 cross and MPV (Figure 5). These genes, which comprised 19.4% and 5.8% of the candidate genes from the liver and breast muscle, were identified as non-additive genes. Thus, most genes in F1 chicken post-hatch breast muscle and liver exhibit additive expression patterns. Genes exhibiting non-additive expression patterns can be further classified into dominance (including enhancing dominance and suppressing dominance), over-dominance, and under-dominance. Specifically, the number of genes showing enhancing dominance, suppressing dominance, over-dominance, and under-dominance in the liver is 538, 429, 1149, and 550, respectively. As for breast muscle, the number of genes exhibiting these non-additive expression patterns is 83, 114,161, and 219, respectively. Overall, these results indicate that additivity is the predominant gene expression pattern in F1 chicken post-hatch breast muscle and liver.

### 3.5. Enrichment Analysis

Subsequently, GO and KEGG enrichment analyses were conducted on the differentially expressed gene sets and gene sets with different expression patterns in the breast muscle and liver tissues using the DAVID online tool. The significantly enriched GO terms and pathways for the seven gene sets (maternal line vs. F1 cross, paternal line vs. F1 cross, paternal line vs. maternal line, enhancing dominance, suppressing dominance, over-dominance, and under-dominance) in the breast muscle (Appendix A) and liver (Appendix A) are provided. Since the breast muscle is a crucial economic trait in chickens for meat production, we further analyzed the BP terms, pathways, and key genes closely related to muscle growth and development in the breast muscle tissue. As shown in Table 1, a total of 11 BP terms closely associated with growth and development were annotated in these seven gene sets mentioned above, such as cardiac muscle contraction, muscle contraction, cell division, skeletal muscle cell differentiation, cell proliferation, skeletal muscle contraction, regulation of ATPase activity, and more. Within these terms, we have identified 61 genes, including *STAT5A*, *TGFB2*, *FGF1*, *IGF2*, *DMA*, *FGF16*, *FGF12*, *STAC3*, *GSK3A*, *GRB2*, and others, which play a pivotal role in muscle growth and development. In addition, a total of six pathways related to growth and development were identified in the breast muscle, including the PPAR signaling pathway, metabolic pathways, TGF-beta signaling pathway, oxidative phosphorylation, and mTOR signaling pathway. Some key genes such as *SLC27A4*, *GLUL*, *TGFB2*, *COX17*, and *GSK3A* were involved (Table 2). Furthermore, to analyze whether the gene sets obtained above are associated with allele-specific functional differences, we compared these seven gene sets with the previously identified sets of genes showing allelic transmission ratio distortion [24] and allele-specific expression [25], but no shared genes were observed.

## 4. Discussion

To date, genome-wide gene expression pattern analysis in chickens has been limited to embryonic brain and liver tissues [23], highlighting a gap in the study of the economic trait and post-hatching stage in chickens. Previously, a full-sib F1 hybrid of meat- and egg-type chickens was established to explore spatiotemporal allele-specific expression and allelic transmission ratio distortion [24,25]. Herein, the previous transcriptome data from day 28 breast muscle and liver samples of the F1 chickens, along with the newly obtained transcriptome data of their parental lines, were further utilized to detect the gene expression patterns of the economic trait (breast muscle) in the post-hatching stage. In addition, due to Chinese consumers’ special dietary habits and culinary preferences, as well as their ability to effectively balance the fast growth performance of meat-type chickens and the superior reproductive performance of egg-type chickens, hybrid broilers have become an important supplementary model in the Chinese meat chicken market [31]. Hence, a better understanding of the inter-group DEGs and gene expression patterns in this F1 population is conducive to revealing the regulation mechanism of muscle growth and development in hybrid broilers. Based on the results of this study, we will further explore the linkages between key genes and phenotypic traits in hybrid broilers with the purpose of providing a theoretical reference for chicken production efficiency.

Among the top 10 genes with the highest expression in the breast muscle and liver, *ENSGALG00010000588*, *COX1*, *COX2*, *ATP6*, and *COX3* were highly expressed in both tissues, while *MYLPF* was only specifically highly expressed in muscle. In accordance with our findings, the top 10 expressed genes in the chicken leg muscle also include the genes *GAPDH*, *COX2*, *COX3*, and *ATP6* [32]. Among these genes, *GAPDH* is one of the most commonly used housekeeping genes [33]. Although the current study shows no significant differences in *GAPDH* gene expression levels among different groups at the same time point, previous spatiotemporal expression profiling analysis of breast and leg muscles has revealed significant variations in the expression levels of this gene across different time points [28,32], indicating that *GAPDH* is not suitable for use as the internal control gene in skeletal muscle. *COX2*, *COX3*, and *ATP6* are genes that encode enzymes involved in processes such as cell energy metabolism and synthesis, respiration, and oxidation, which are crucial for maintaining cellular functions and energy production [34,35,36]. Similarly, the *MYLPF* gene, which encodes the fast myosin regulatory light chain, was found to be highly expressed in muscle tissues such as the abdominal muscle, longissimus dorsi, and gastrocnemius in goats [37]. Additionally, polymorphisms in the regulatory region of the porcine *MYLPF* gene were identified to be associated with meat quality traits [38]. Therefore, these top ten genes are essential for maintaining the vital activities of the cell and might play crucial roles in the growth and development processes of chickens.

Gene expression serves as an intermediate phenotype between genotype and specific traits [15]. DEGs related to muscle growth and development can be considered the primary drivers of genetic variation in chicken growth. Our study revealed that the DEGs in the comparison of the paternal line vs. maternal line in breast muscle were significantly lower than those in the comparisons of the maternal line vs. F1 cross and the paternal line vs. F1 cross, with a similar trend observed in the liver tissue, indicating a significant shift in regulatory mechanisms between the parental lines and the F1 chickens. Moreover, studies on genome-wide gene expression patterns have shown that the predominant gene expression pattern varies across species, traits, and generations, as evidenced in maize (78% genes with additivity), rice (50% genes with additivity), pufferfish (4.6% genes with additivity), and cyprinidae (F1: 23.6% genes with additivity; F2: 10% genes with additivity) [19,39,40,41]. Due to the spatiotemporal specificity of gene expression, we investigated the gene expression patterns of the economic traits in post-hatch chickens. In this study, 80.6% of genes in the liver and 94.2% of genes in the breast muscle exhibited additive expression, which is consistent with the predominant gene expression pattern observed in the embryonic brain (73.1% genes with additivity) and liver (93.2% genes with additivity) [23]. These findings suggest that additivity is the predominant gene expression pattern in chickens across stages and tissues.

GO analysis revealed that the DEGs and non-additive genes are primarily involved in processes related to cell growth, muscle development, and various cellular activities including lipid transport, migration, triglyceride homeostasis, smooth muscle contraction, positive regulation of cell proliferation, and cell death. Of all the DEGs and non-additive genes identified in the significantly enriched GO BP terms involved in muscle growth and development, some were previously reported to be closely related to growth and development, such as *STAT5A*, *TGFB2*, *FGF1*, *IGF2*, *DMA*, *FGF16*, *FGF12*, *STAC3*, *GSK3A*, and *GRB2*. Among these, *STAT5A* has been linked to cell specification, proliferation, differentiation, and survival at the molecular level [42]. Additionally, *STAT5A* has been found to play a role in immunity and growth by positively regulating porcine *CISH* gene transcription, and the SNP (g.566C > T) of *STAT5A* was associated with the piglet growth trait [43]. TGFB2 is a critical ligand in the TGF-β signaling pathway [44]. Knockout of *TGFB2* resulted in organ dysfunction affecting the heart, lung, skeletal muscle, urogenital tract, inner ear, and eyes in mice [45]. In chickens, analysis of the embryonic expression of TGF-β ligand and receptor genes reveals that *TGFB2* exhibits dynamic and overlapping patterns in numerous embryonic cell layers and structures, playing a role in specific developmental processes such as somitogenesis, cardiogenesis, and vasculogenesis [44]. In addition, the *TGFB2* T (−640) > C SNP was associated with growth and body composition traits in a broiler chicken [46]. As a member of the fibroblast growth factor (*FGF*) gene family, *FGF1* plays a role in adipocyte differentiation and adipose tissue remodeling [47]. In chickens, *FGF1* gene expression is positively correlated with intramuscular fat content in male thigh muscle but showed a negative correlation in female birds [48]. *IGF2* has been associated with a desirable reduction in body fat depth and a desirable increase in the intramuscular fat content of pigs [49]. Recent studies have shown that the *IGF2* gene is closely related to chicken growth, carcass, and meat quality traits, such as muscle growth and fat metabolism [50,51]. The *DMA* (DM α chain) gene, as part of the non-classical MHC class II, is essential for antigen presentation through the production of the DM protein, and its polymorphism has been associated with disease resistance traits in chickens, including total IgY concentration and Newcastle disease antibody titers [52].

For significantly enriched non-additive effect key genes, *FGF12* in geese could regulate the cell cycle gene expressions of *CCND1*, *CCNA2*, *MAD2*, and *CHK1* and inhibit follicular granulosa cell apoptosis through ERK phosphorylation [53]. *FGF16*, as an adipogenic factor, is expressed abundantly in brown adipose tissue. In addition, *FGF16* can promote goat intramuscular preadipocyte differentiation and triglyceride synthesis via *FGFR4* [54]. As a member of the *STAC* family, the *STAC3* gene is specifically expressed and negatively regulates satellite cell differentiation in chicken skeletal muscle [55]. Similarly, the *STAC3* gene has also been demonstrated to be essential for the development and function of skeletal muscle in mice [56]. *GSK3* is a serine–threonine kinase with two isoforms, namely *GSK3A* and *GSK3B*. The *GSK3A* could affect sperm motility and acrosome reaction, due to the regulation of energy metabolism [57]. *GRB2* was initially discovered for its role in cell proliferation, differentiation, and survival [58]. Further study found that *GRB2*-deficient mouse T cells exhibit developmental defects and increased differentiation of Th1 and Th17 cells [59].

Moreover, chicken muscle growth and development are intricate processes influenced by a multitude of genes and regulated through various pathways. KEGG analysis showed that six significantly enriched pathways related to muscle growth and development were identified, namely the PPAR signaling pathway (maternal line vs. F1 cross), the PPAR signaling pathway (paternal line vs. F1 cross), metabolic pathways (paternal line vs. maternal line), the TGF-beta signaling pathway (suppressing dominance), oxidative phosphorylation (under-dominance), and the mTOR signaling pathway (under-dominance). Several key genes were identified, including *SLC27A4*, *GLUL*, *TGFB2*, *COX17*, and *GSK3A*. The PPAR signaling pathway is associated with lipid deposition in chicken breast muscle tissue. After activating PPARG, a transcription factor in the PPARG signaling pathway, the expression of lipogenesis genes (*LPL*, *SCD,* and *CD36*) was upregulated to promote TG synthesis [60]. Oxidative phosphorylation is a pathway that uses energy released from the oxidation of nutrients to produce ATP [61]. A recent study showed the downregulation of differentially expressed proteins associated with oxidative phosphorylation, indicating a high level of energy metabolism in muscle [62]. The TGF-beta signaling pathway is involved in cell proliferation, differentiation, and signal transduction [63]. The inhibition of the TGF-beta signaling pathway in vivo could downregulate the expression of *DAZL*, *CVH*, and *CKIT* genes and reduce germ cell formation [64]. The mTOR signaling pathway is closely linked to the development of chicken granulosa cells, and its activation positively regulates granulosa cell proliferation, leading to an increase in the number of developing follicles [65]. For the key genes in these pathways, *SLC27A4* is recognized as a potential candidate gene for traits associated with fat deposition, and its polymorphisms correlate with backfat thickness and body weight at birth in pigs [66]. As a member of the glutamine synthetase family, *GLUL* plays an important role in the formation of meat flavor substances and muscle nutrition [67]. *COX17* knockout mice die early in embryonic development due to severely reduced CCO activity [68]. Furthermore, the gene functions of *TGFB2* and *GSK3A* align with the relevant descriptions in the GO analysis section.

## 5. Conclusions

Collectively, this is the first report of gene expression patterns in the chicken post-hatch economic trait. The current study indicates that in chickens, whether during the embryonic period or post-hatching, the predominant gene expression pattern in multiple tissues shows additive expression. A large number of DEGs were identified among different groups, and several gene sets with non-additive expression patterns were obtained. Furthermore, enrichment analysis revealed that the DEGs and non-additive genes found in significantly enriched BP or pathways were determined to be closely related to the growth and development of breast muscle. In the future, conducting functional and association analyses of these identified DEGs and non-additive genes will be crucial for exploring the genetic basis of economically important traits in crossbreeding of domestic animals.

## Figures and Tables

**Figure 1 animals-14-01335-f001:**
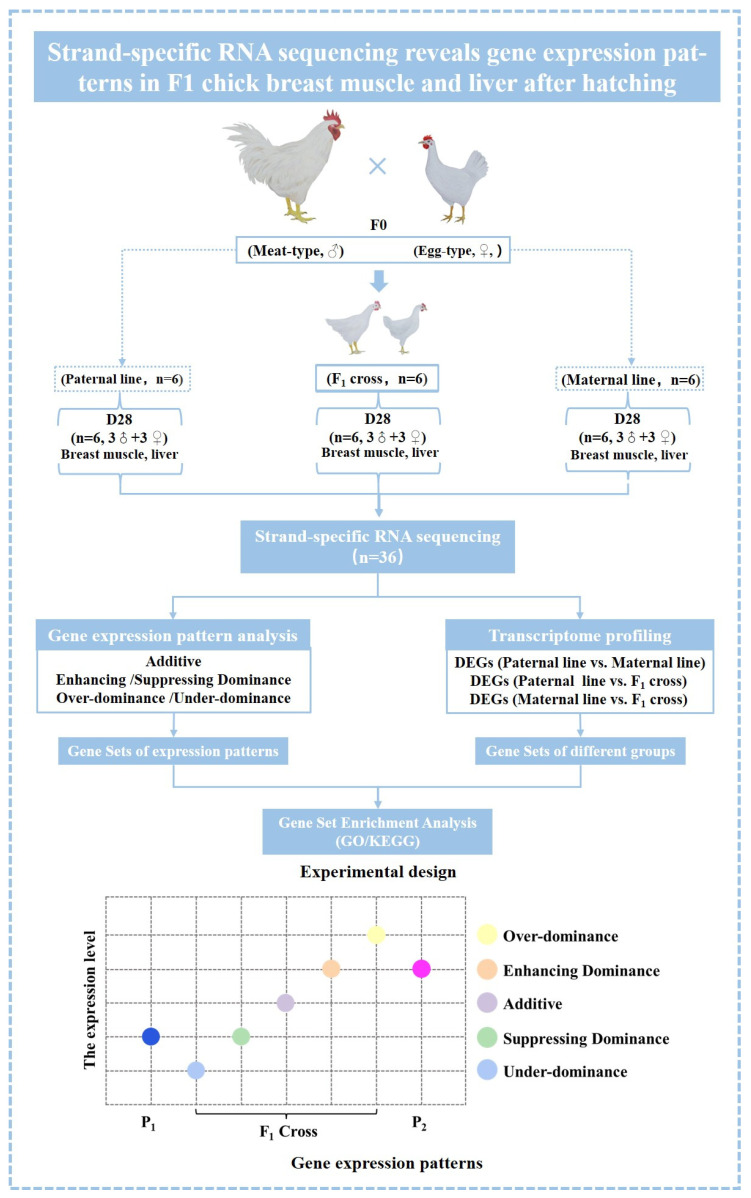
Experimental design and gene expression patterns. Meat-type: Recessive White chicken; Egg-type: Lohmann pink layer; F1 cross: Recessive White chicken × Lohmann pink layer cross; P1: Parent 1; P2: Parent 2; D28: 28-day-old.

**Figure 2 animals-14-01335-f002:**
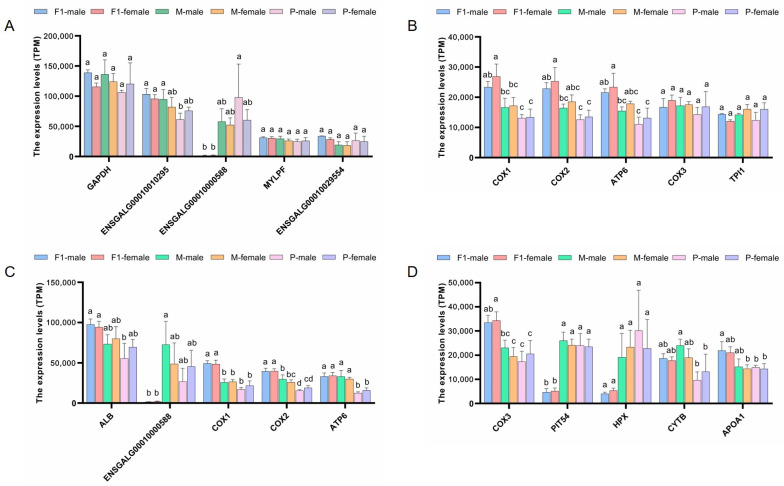
Top ten genes with the highest mRNA levels in breast muscle (**A**,**B**) and liver (**C**,**D**). All results are displayed as mean ± SEM. *n* = 3. ^a–c^ *p* < 0.05.

**Figure 3 animals-14-01335-f003:**
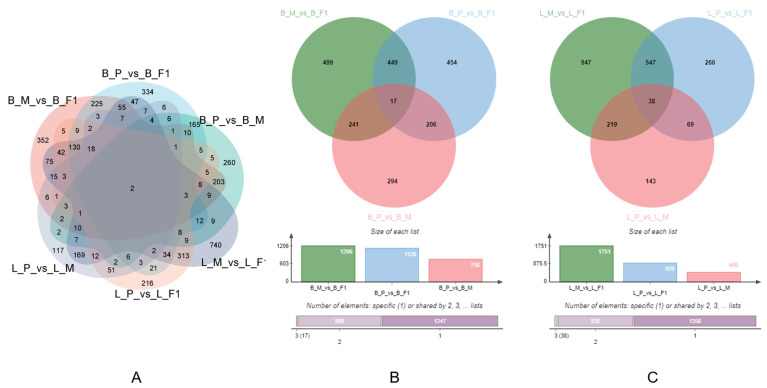
The number of DEGs in all groups (**A**), breast muscle (**B**), and liver (**C**). B_M_vs_B_F1: breast muscle, maternal line vs. F1 cross; B_P_vs_B_F1: breast muscle, paternal line vs. F1 cross; B_P_vs_B_M: breast muscle, paternal line vs. maternal line; L_M_vs_L_F1: liver, maternal line vs. F1 cross; L_P_vs_L_F1: liver, paternal line vs. F1 cross; L_P_vs_L_M: liver, paternal line vs. maternal line.

**Figure 4 animals-14-01335-f004:**
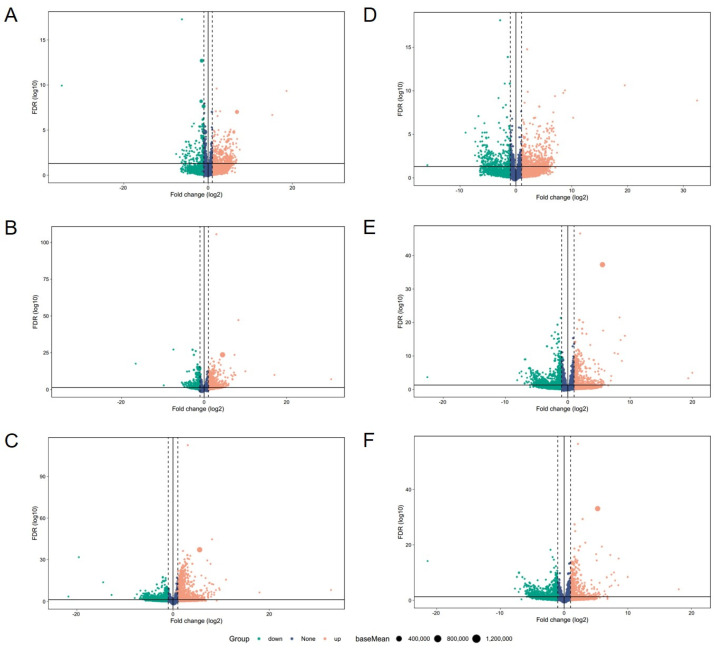
DEGs identified between different groups. (**A**) Liver, paternal line vs. maternal line. (**B**) Liver, paternal line vs. F1 cross. (**C**) Liver, maternal line vs. F1 cross. (**D**) Breast muscle, paternal line vs. maternal line. (**E**) Breast muscle, paternal line vs. F1 cross. (**F**) Breast muscle, maternal line vs. F1 cross.

**Figure 5 animals-14-01335-f005:**
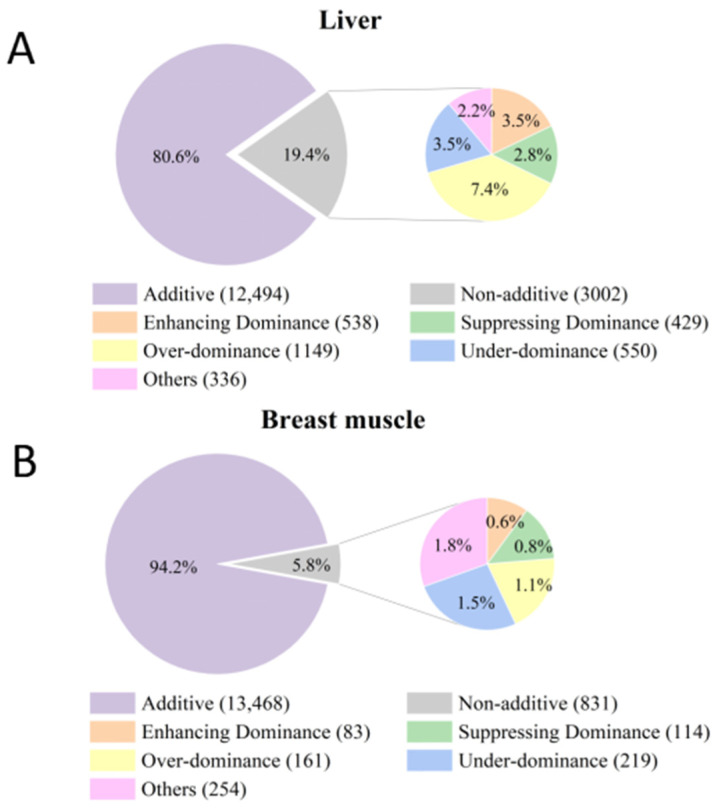
Gene expression patterns in Liver (**A**) and breast muscle (**B**).

**Table 1 animals-14-01335-t001:** Significantly enriched BP terms involved in muscle growth and development.

Gene Set	Term ID	Term	*p*-Value	Genes
Maternal line vs. F1 cross	GO:0006869	lipid transport	0.017905476	*ENSGALG00010018145*, *APOA1*, *APOA4*, *ABCA3*, *GM2A*, *SPNS2*, *ABCA5*
	GO:0070328	triglyceride homeostasis	0.025746286	*LPL*, *RORA*, *APOA1*, *HNF4A*
	GO:0006939	smooth muscle contraction	0.032805327	*ROCK2*, *CHRNB4*, *FKBP1B*, *HTR7*
Paternal line vs. F1 cross	GO:0008284	positive regulation of cell proliferation	0.004132833	*STAT5A*, *ERBB2*, *EXFABP*, *COPS9*, *MEIS2*, *S100B*, *MARCKSL1*, *TGFA*, *TGFB2*, *BAMBI*, *PRLR*, *FGF1*, *CDCA7L*, *CNOT6L*, *INSR*, *TBX6*, *HSP90AA1*, *IGF2*
	GO:0008219	cell death	0.020674525	*BRINP1*, *CATH1*, *CTSD*, *TGFB2*
Paternal line vs. maternal line	GO:0051482	positive regulation of cytosolic calcium ion concentration involved in phospholipase C-activating G-protein coupled signaling pathway	0.003182625	*GPR55*, *LPAR4*, *DRD3*, *GPR65*, *LPAR6*
	GO:0019886	antigen processing and presentation of exogenous peptide antigen via MHC class II	0.011956304	*IGLL1*, *CD74*, *DMB2*, *DMA*
Suppressing dominance	GO:0008284	positive regulation of cell proliferation	8.83 × 10^−4^	*BAMBI*, *GREM1*, *PRLR*, *FGF16*, *FGF12*, *TGFB2*
Under-dominance	GO:0003009	skeletal muscle contraction	0.005685356	*TNNC2*, *STAC3*, *TNNC1*
	GO:0070507	regulation of microtubule cytoskeleton organization	0.022384332	*GSK3A*, *TRAF3IP1*, *RHOA*
	GO:0008286	insulin receptor signaling pathway	0.029955405	*GSK3A*, *GRB2*, *EIF4EBP2*

**Table 2 animals-14-01335-t002:** Significantly enriched pathways related to muscle growth and development.

Gene Set	Pathway	*p*-Value	Genes
Maternal line vs. F1 cross	PPAR signaling pathway	0.046223775	*FABP6*, *LPL*, *HMGCS2*, *APOA1*, *FABP1*, *ACOX2*, *EHHADH*, *PLIN1*, *CD36*
Paternal line vs. F1 cross	PPAR signaling pathway	0.010920518	*FABP6*, *SLC27A1*, *SCD5*, *CYP27A1*, *FABP1*, *FABP7*, *SCD*, *ACSBG2*, *EHHADH*, *SLC27A4*
Paternal line vs. maternal line	Metabolic pathways	0.005143418	*CYP2W1*, *HAO1*, *ENSGALG00010028858*, *CTH*, *CYP27A1*, *SHPK*, *SMPD3*, *FLAD1*, *IL4I1*, *KHK*, *CMPK2*, *GLUL*, *KYNU*, *GYS2*, *BTD*, *CA3A*, *CHIA*, *GATM*, *SGPP2*, *ENSGALG00010004334*, *TBXAS1*, *GCDH*, *ACOX2*, *GPX2*, *MAN1A1*, *DCT, AOC1*, *UGT8*, *ENSGALG00010029214*, *ACACB*, *ENSGALG00010011814*, *B3GNT2*, *UROC1*, *SMYD2*, *SGMS1*, *GLYCTK*, *ENSGALG00010020715*, *HKDC1*, *PDE10A*, *PYCR1*, *MGAT5B*, *VNN2*, *CNDP1*, *PIPOX*, *ENSGALG00010011927*, *INPP5J*, *PPOX*, *ENSGALG00010017668*, *ADCY7, AKR1D1*, *CDA*, *ADH6*, *PTGS2*, *HAAO*, *DEGS2*, *SCD5*, *CHKA*, *PLD4*, *RFKL*, *ALDH8A1*, *CSGALNACT1*, *ASNS*, *CYP21A1*, *ENSGALG00010011278*, *PAH*, *HAO2*
Suppressing dominance	TGF-beta signaling pathway	0.025495457	*BAMBI*, *GREM1*, *ID3*, *TGFB2*
Under-dominance	Oxidative phosphorylation	1.10 × 10^−6^	*ENSGALG00010027713*, *SDHB*, *NDUFAB1*, *NDUFA12*, *COX17*, *NDUFB4*, *NDUFA13*, *ATP5E*, *COX6A1*, *NDUFB3*, *NDUFS5*
	mTOR signaling pathway	0.048844595	*GSK3A*, *EIF4B*, *RHOA*, *LAMTOR2, GRB2*, *HRAS*

## Data Availability

The data presented in this study are available on request from the corresponding author. The data are not publicly available due to the subsequent selection of candidate genes for functional validation and correlation analysis is still pending completion.

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
