# Peer review of "Strand-Specific RNA Sequencing Reveals Gene Expression Patterns in F1 Chick Breast Muscle and Liver after Hatching"

_animals, 2024, doi:10.3390/ani14091335_

Round 1

Reviewer 1 Report

Comments and Suggestions for Authors

Poultry farming is the most important area of agriculture, meeting the population's needs for protein of animal origin. Scientific research makes it possible to increase the efficiency of this industry by increasing poultry meat productivity. This study examines gene expression in two major meat poultry products, breast muscle and liver. Selection of meat poultry is carried out in the direction of increasing the mass of the pectoral muscle. The authors conducted a large amount of research and obtained interesting results.

Remarks: Is the number of samples being tested sufficient? Perhaps their number needs to be increased for a more reliable result. And most importantly: why didn’t the authors conduct a parallel study of the size and mass of the pectoral muscle of the birds they studied? There is no point in studying gene expression if expression is not linked to phenotypic data. That is, the most important aspect of such a study will be the influence of the level of expression of certain genes on the mass and size of the pectoral muscle of chickens. This part of the work is missing. Therefore, the authors may annotate this study as a pilot study.

Author Response

Dear Editors:  

We truly appreciate the reviewer’s valuable comments and helpful suggestions. We have revised our manuscript according to your suggestions. All changes made to the manuscript are shown in red color. We hope that the revised manuscript will meet your requirement now. Below, there are our point-by-point responses to the reviewer’s comments/ questions. Please contact me if you have any other questions. I quite appreciate it for your help with our manuscript. Thank you so much!

Yours sincerely,

Peng Ren

April 25, 2024

Additive list

To reviewer#1:

Is the number of samples being tested sufficient? Perhaps their number needs to be increased for a more reliable result.

au: We appreciate your suggestion regarding the sample size for testing. In the transcriptome analysis of the F1 hybrid population in our previous study, we randomly selected 3 males and 3 females from the F1 population at 1, 28, and 56 days of age, respectively, for transcriptome analysis. Building upon the previous transcriptome sequencing data of the breast muscle and liver tissues of the F1 population at 28 days of age, this study additionally sequenced the transcriptomes of 3 males and 3 females from both pure paternal and maternal lines to explore the gene expression patterns related to the economic traits of this hybrid population. While increasing the number of samples could potentially enhance the reliability of the results, our current sample size was determined based on the specific research objective.

Why didn’t the authors conduct a parallel study of the size and mass of the pectoral muscle of the birds they studied? There is no point in studying gene expression if expression is not linked to phenotypic data. That is, the most important aspect of such a study will be the influence of the level of expression of certain genes on the mass and size of the pectoral muscle of chickens. This part of the work is missing. Therefore, the authors may annotate this study as a pilot study.

au: Thank you for your valuable suggestion. In this study, our primary objective was to conduct an initial exploration of gene expression patterns and key genes in the breast muscle of F1 chicks post-hatching. We acknowledge your point and intend to further investigate the correlations between these key genes and phenotypic traits, aiming to elucidate the mechanisms influencing the growth and development of broiler hybrid chickens. We appreciate your feedback and will work towards enhancing the study by incorporating the suggested aspects.

Reviewer 2 Report

Comments and Suggestions for Authors

Heterosis has been widely utilized in crossbreeding programs for animals and crops, yet the molecular biology underlying this phenomenon remain enigmatic. Understanding the post-hatch gene expression patterns is pivotal for exploring the genetic basis underlying economically important traits in crossbreeding of chickens. In the current study, Zhao et al. and their colleagues have conducted gene expression analysis on F1 chicken breast muscle and liver tissues by ssRNA-seq at 28 days and revealed additivity as the predominant gene expression pattern in post-hatch muscle and liver. The findings provide valuable insights for domestic animal crossbreeding. The experiment was well designed and well written. The following revision could improve the quality of this paper.

L85, please provide the ethics certification and the relative code for the animal protocol.

L89, why the tissue samples from day 28 of F1 population has been chosen?

L123, 2.4. Gene expression pattern analysis, please cite the right references for this analysis.

L150, please provide the quality control data as supplemental files for the omics analysis, including the precision and accuracy of sequencing results.

Figure 2 and others, the abbreviations of the genes of the mRNA expression data should be written in italic.

Figure 3, the resolution of the figure 4 needs to be improved; and checking the similar issues for other figures.

Please do the q-PCR analysis to check some of the key DEGs. This is very important to check the accuracy and reliability of the omics data.

Checking the writing, such as space need to be added before and after “=”; P value need to be written in italic; abbreviation of the full name should be written when it was firstly appeared in the paper.

Table 2, please correct “under-dominance” to “Under-dominance”; and revising the similar issues throughout the paper.

The gramma and language need to be improved by a native English speaker.

Author Response

Dear Editors:  

We truly appreciate the reviewer’s valuable comments and helpful suggestions. We have revised our manuscript according to your suggestions. All changes made to the manuscript are shown in red color. We hope that the revised manuscript will meet your requirement now. Below, there are our point-by-point responses to the reviewer’s comments/ questions. Please contact me if you have any other questions. I quite appreciate it for your help with our manuscript. Thank you so much!

Yours sincerely,

Peng Ren

April 25, 2024

Additive list

To reviewer#2:

L85: please provide the ethics certification and the relative code for the animal protocol.

au: Thanks for your comment. The ethics certification and the relative code for the animal protocol was shown in Line 417.

L89: why the tissue samples from day 28 of F1 population has been chosen?

au: Based on previous studies on growth rate and transcriptome sequencing data, we observed a significant turning point and differences in the growth rate phenotype and number of differentially expressed genes of the hybrid F1 population at 28 days compared to other time points. Therefore, in this study, we selected the time point of 28 days to further analyze the gene expression patterns of this hybrid population. The rationale for selecting this time point is described in Line 90.

L123: 2.4. Gene expression pattern analysis, please cite the right references for this analysis.

au: Thanks for your suggestion. We have selected a more appropriate reference for categorizing gene expression patterns, Line 484.

L150: please provide the quality control data as supplemental files for the omics analysis, including the precision and accuracy of sequencing results.

au: The quality control data of the F1 transcriptome data was showed in our previous study [26] and the quality control data of the newly sequenced transcriptome data of parental lines was added in Table S1, Line 157, 626.

Figure 2 and others, the abbreviations of the genes of the mRNA expression data should be written in italic.

au: Thanks for your careful checks. We have italicized all gene names throughout the entire manuscript.

Figure 3, the resolution of the figure 4 needs to be improved; and checking the similar issues for other figures.

au: Thanks for your valuable suggestions. We checked and optimized the resolution of all images and data to ensure compliance with publication requirements.

Please do the q-PCR analysis to check some of the key DEGs. This is very important to check the accuracy and reliability of the omics data.

au: Thank you very much for your suggestion regarding the validation of some key genes. In this study, we utilized strand-specific RNA sequencing technology to enhance the accuracy of the sequencing data results. However, as the corresponding tissue samples for sequencing have been destroyed due to storage expiration, we are unable to proceed with q-PCR validation. We will take note of this issue for future research projects.

Checking the writing, such as space need to be added before and after “=”; P value need to be written in italic; abbreviation of the full name should be written when it was firstly appeared in the paper.

au: Thanks a lot. Upon closer inspection, we did find some formatting issues in the manuscript. We have made revisions according to your helpful suggestion.

Table 2, please correct “under-dominance” to “Under-dominance”; and revising the similar issues throughout the paper.

au: Thanks for your suggestion. We have revised these issues throughout the paper.

The gramma and language need to be improved by a native English speaker.

au: Thanks for your comments. According to your advice, the manuscript has been thoroughly reviewed and edited by proficient native English speakers to enhance language accuracy, grammar, punctuation, spelling, and overall style.